# The Method for Risk Evaluation in Assembly Process based on the Discrete-Time SIRS Epidemic Model and Information Entropy

**DOI:** 10.3390/e21111029

**Published:** 2019-10-24

**Authors:** Mengyao Wu, Wei Dai, Zhiyuan Lu, Yu Zhao, Meiqing Wang

**Affiliations:** 1School of Reliability and Systems Engineering, Beihang University, Beijing 100191, China; wmy0317@buaa.edu.cn (M.W.); zhaoyu@buaa.edu.cn (Y.Z.); 2School of Mechanical Engineering and Automation, Beihang University, Beijing 100191, China; rselzy@163.com (Z.L.); wangmq@buaa.edu.cn (M.W.)

**Keywords:** discrete SIRS epidemic model, optimal assembly path, assembly complexity, risk evaluation, information entropy

## Abstract

In the past decade years, much attention has been attached on assembly process reliability in manufacturing system, because the quality and cost of product are highly determined by assembly process. However, existing research on reliability in assembly are mainly focused on study of size deviation propagation. In this paper, the method for risk evaluation in assembly process based on the discrete-time SIRS epidemic model and information entropy was proposed. Firstly, aiming at the issue of assembly process optimization, innovative solutions are proposed from the perspectives of reliability and cost by decomposing the assembly into general path and rework path. Secondly, the propagation mechanism of defects in optimal assembly approach were studied through combining the infectious disease model and information entropy. According to the bifurcation phenomenon in the SIRS model, the entropy increment of assembly process $\Delta H_{base}$ when defect emergence occurs is calculated. Thirdly, the information entropy increment of optimal assembly approach ΔH is used to evaluate the assembly risk by comparing with the $\Delta H_{base}$. Finally, a case study of assembly risk evaluation for the oil pump was presented to verify the advantage of this method.

## 1. Introduction

With the development of manufacturing industry and increasing customer’s demand, manufacturers are facing the challenge of improving the reliability and diversity of products [1,2]. Assembly process is one of the most important parts during the manufacturing cycle and the quality of products is highly influenced by the assembly process due to the complexity in assembly manufacturing system [3,4]. Latent defects in assembly process are easily triggered to dominant defects when environment conditions change, which could lead to the phenomenon that defect emergence often breaks out. Therefore, significant attention has been paid to the assembly process under the manufacturing environment.

At present, the research on the complexity in assembly manufacturing system mainly focuses on structure complexity, process complexity, and control complexity [5,6,7,8,9], as Figure 1 shows. Many scholars have done considerable work on assembly system. Kusiak and He [10] put forward the concept of agile assembly, and gave three rules applicable to support the design of products. Heilala and Voho [11] showed how to create flexible capability and capacity in the final assembly systems. To meet the changing customers’ demands, the problem for reconfiguring flexible assembly line systems was solved through the application of motion genes [12]. Xu and Liang [13] proposed an integrated approach for product module selection and assembly line design/reconfiguration problems. The quality loss functions were used to quantify noncomparable and possibly conflicting performance criteria in their study. In the research by Bryan et al. [14], an innovative method for the concurrent design of a product portfolio and its corresponding assembly system was presented, which could lead to the minimum of oversupply in differentiating modules and the maximum of the efficiency in the assembly line. There are two main problems in mixed model assembly lines, one is sequencing of different models, and the other is balancing of assembly line. Saif et al. [15] proposed the multi-objective artificial bee colony algorithm for simultaneous sequencing and balancing of mixed model assembly line to overcome these problems.

It is necessary to assess the reliability of one assembly system since assembly system is so complicated. Nowadays, the method of risk evaluation develops fast. Failure mode and effects analysis (FMEA) and stream of variation (SOV) are two of the most important risk evaluation tools that have been used widely in many fields these years. However, the traditional FMEA is very subjective, because it relies on people’s experience much. Many scholars has criticized and improved it in theirs research [16,17,18,19]. Fattahi and Khalilzadeh [20] presented a novel hybrid method to evaluate various failure modes that are based on FMEA, and extended MULTIMOORA and AHP methods under fuzzy environment. Sankar and Prabhu [21] put up with a modified approach for prioritization of failures in a system FMEA. Tooranloo and Ayatollah [22] proposed an innovative model for FMEA that is based on the intuitionistic fuzzy approach. There are significant literature referring to the variation propagation during the assembly process [23,24,25,26,27]. The dimensional variation will be introduced to each assembly process, and further influence the assembly quality [26]. Ceglarek et al. [23] discussed the concept of time-based competition in manufacturing and design based on a review of on-going research related to SOV methodology. Camelio et al. [28] developed a methodology to assess the dimensional variation propagation in a multi-station compliant assembly system based on linear mechanics and a state space representation. Zhou et al. [29] took the different motion vector, which is a concept from the robotics field, to state the geometric deviation of the workpiece. The model that they put up had potential to be applied in complicated machining processes. As we all know, entropy can be used as a measurement of the uncertainty or information content of a random event. To an assembly system, it will become more chaotic when defects occur. Therefore, it is feasible to use entropy for the analysis of assembly system. Li et al. [30] improved the multi-source uncertainty method during the assembly process based on surrogate model and information entropy. Liu et al. [31] defined the welding system complexity through information entropy. Fujimoto and Ahmed [32] proposed a new evaluation approach for measuring the complexity of an assembly system by applying the information entropy. Barchielli et al. [33] introduced a new information relative entropy formulation to measurement uncertainty relations.

In the practice assembly process, the location of inspection station will lead to the difference of rework path. Therefore, there are plenty of assembly schemes to an assembly system. How to choose the optimal assembly scheme has become one of the most important problems that need to be solved. As we all know, most scheduling problems are typical NP problems, and many scholars have conducted corresponding studies on this issue. Inspired by natural phenomenon and intelligent methods, Glover and Greenberg [34] put up with a new algorithm that effectively solves scheduling problems and other combinatorial optimization problems. In recent years, various scheduling optimization algorithms have been widely applied in production scheduling problems to improve the efficiency of assembly process. For assembly-line scheduling problem, there are swarm optimization algorithm [35], genetic algorithm [36], migratory bird optimization algorithm [37], hybrid algorithm [38,39], and so on. As for the current popular multi-variety and small-batch personalized customized production process, scheduling optimization is more widely applied and the corresponding research results are quite abundant [40,41,42,43]. However, these traditional scheduling methods consider separately processing and assembly, and the product is divided into multi jobs, the constraint relationships with the process are ignored. Aiming at solving this problem, the integrated scheduling method of complex products comes into being. This integrated scheduling algorithm simultaneously scheduled the processing and assembly, which improved the degree of parallel production and saved time [44].

SIRS model is a typical classical model that has been widely applied to research the disease propagation among populations. The stochastic SIRS epidemic model with a non-linear incidence rate in the population was studied and proposed in many literature [45,46,47,48]. Hu et al. [49] discussed the dynamical behaviors of a class of discrete-time SIRS epidemic model. Cai et al. [50] extended a classical SIRS epidemic model with the infectious forces through introducing random fluctuations. According to our previous research [51], there are plenty of similarities between the defects emergence in assembly process and disease propagation in population. We have had a comparison from the aspects of infectious source, infectious path, and susceptible individuals, just as Table 1 shows. We apply the SIRS epidemic model to simulate the critical entropy during the assembly process when defects emerge. The existing literature on risk evaluation in assembly system is quite comprehensive. However, the idea for risk evaluation in the assembly process we proposed is quite different from the previous research.

The remainder of this paper is organized as follows. Section 2 develops an innovative approach for choosing the optimal assembly path while considering the structure complexity. The risk evaluation method based on SIRS epidemic model and information entropy is proposed to solve the dynamic complexity of assembly system in Section 3. Section 4 takes the assembly process of oil pump as an example for verifying the proposed method. Section 5 provides the conclusions of this work and discussions. The framework is shown as Figure 2.

**Assumptions**: As risk is a common phenomenon, it occurs in all areas of manufacturing. There are various scientific disciplines that deal with risk analysis, e.g., probability calculus, statistics, econometrics, image recognition theory, reliability theory, operational research, theory of organization and management, etc. [53].

There are some assumptions involved in the method that we proposed.

***Assumption 1***: The reliability of each assembly stations is equal and constant.

***Assumption 2***: Different production lines are independent of each other.

***Assumption 3***: Products in assembly process are divided into three compartments according to their health states. In a certain condition, three states can transform into each other.

## 2. Optimal Assembly Path Selection based on Reliability and Cost

In the manufacturing system, each assembly line contains a series assembly processes, and each process needs to perform specific assembly functions. Mass production is always involved in modern manufacturing. The flexibility and efficiency of assembly can be improved through using multiple production lines. In addition, it is necessary to set up appropriate inspection stations in proper position to guarantee the assembly quality and prevent the appearance of unqualified products that are caused by the previous assembly process. Qualified products shall be released for subsequent assembly, and unqualified products shall be returned to the previous procedure for maintenance. However, the repair path in practice the assembly process is not unique because of the fault tolerance and repair of some process and the diversity of inspection stations. The assembly system is complicated enough when the assembly structure is considered. From the perspective of production management and decision-making, factors such as time, cost, and reliability need to be comprehensively considered to determine the optimal inspection station position and rework path.

Actually, rework products will further affect the output capacity of the assembly system. It is of great importance to consider the rework path when building the assembly model. Based on the graphical method and structure complexity, the assembly system is first transformed into assembly network, just as Figure 3 shows.

In the model that we built, the solid arrows indicate the assembly process; The dashed arrows indicate the rework path; Black solid circles indicate the inspection stations for raw materials, and final products; Light colored solid circles indicate process inspection station; The empty circles indicate the buffer zones. The reliability of each process is assumed to be p and the maximum assembly capacity of process is M.

As Figure 3 shows, suppose that the inspection process is set after the assembly process $P_{i}$. The assembly model can be decomposed, as in Figure 4. In the decomposition model, the assembly line consists of general line and rework line, which represent assembly process of normal products and repaired products, respectively.

In the general line, the assembly capacity can be described as

(1)$$O_{j}(G)=I_{j}p^{n}$$

Additionally, the assembly capacity in rework line is 

(2)$$O_{j}(R)=I_{j}p^{S_{ij}-1}qp^{t_{ij}}=I_{j}p^{S_{ij}+t_{ij}-1}q$$

Here, q=1−p, i=1,2,…,n, and j=1,2,…,m. $S_{ij}$ is the sequence number of inspection station and $t_{ij}$ is the number of processes behind the repaired station.

Therefore, the quantity of products assembled in production line j is 

(3)$$O_{j}=I_{j}p^{n}+I_{j}p^{S_{ij}+t_{ij}-1}q$$

According to the barrel effect principle, the minimum assembly capacity process determines the maximum assembly capacity of process line. Therefore, the maximum assembly capacity of line j is

(4)$$O_{j\max}=\min\{M_{ij}(p^{r_{ij}+1}+\alpha_{ij}p^{r_{ij}+k+1}q)\}$$

Here, $r_{ij}$ is quantity of processes behind process $P_{ij}$ in assembly line and k represents the number of repaired station. Meanwhile, $\alpha_{ij}=\{\begin{array}{c} 0,P_{ij}\in reworkline\\ 1,P_{ij}\notin reworkline \end{array}$.

Further, the maximum assembly capacity of system can be obtained, as follows:(5)$$O_{\max}=\sum_{j=1}^{m}O_{j\max}$$

For a given assembly task D, the condition that $D\leq O_{\max}$ has to be met. Here, $D=(d_{1},d_{2},\ldots ,d_{m})$. The input of each assembly line is 

(6)$$I_{j}=d_{j}/(p^{n}+p^{S_{ij}+t_{ij}-1}q)$$

The input quantity of each process in general line is 

(7)$$Q_{ij}(G)=I_{j}p^{V_{ij}-1}$$

Here, $V_{ij}$ is the sequence number of $P_{ij}$ in the assembly line.

The input quantity of each process in rework line is 

(8)$$Q_{ij}(R)=I_{j}p^{V_{ij}+k-1}$$

Therefore, the actual input loading of each process is 

(9)$$l_{ij}=Q_{ij}(G)+\beta_{ij}Q_{ij}(R)$$

Here, $\beta_{ij}=\{\begin{array}{c} 1,P_{ij}\in reworkline\\ 0,P_{ij}\notin reworkline \end{array}$.

We can get the loading vector of assembly system, as follows:(10)$$L_{j}=(l_{1j},l_{2j},\ldots ,l_{nj})$$

For each assembly process, we need to find the minimum assembly capacity $c_{ij}$, and the minimum capacity vector is $C_{j}=(c_{1j},c_{2j},\ldots ,c_{nj})$.

The reliability of whole system is 

(11)$$R=\sum_{j=1}^{m}\left\{\mathrm{Pr}(X\geq C_{j})-\mathrm{Pr}(\overset{m-1}{\underset{j=1}{\cup }}\left\{X\geq C^{i,j}\right\})\right\}$$

Here, $C^{i,j}=C_{i}⊕C_{j}$.

## 3. Risk Evaluation of Assembly Process

Based on the method that is proposed above, an optimal assembly method considering reliability and cost can be obtained. However, some latent defects will be excited in the assembly process because of some external stress. According to our previous research [51,52], we compared the similarities between the defects’ propagation in assembly process and the spread of pathogens in population, and put up with the creative idea that using the SIRS epidemic model to research the assembly process. Products in the assembly process are divided into three states, that is, susceptible state, infectious state, and recovery state, as Figure 5 shows.

### 3.1. Discrete-Time SIRS Epidemic Model in Assembly Process

According to the SIRS model in Figure 5, choose a time step size Δt>0. Suppose that $S_{t}=S(t),S_{t+1}=S(t+\Delta t)$ is always true for any t>0. Therefore, when Δt is small enough, we will have the following equations:(12)$$\{\begin{array}{c} S^{\prime }=\underset{\Delta t\to 0}{\lim}\frac{S_{t+1}-S_{t}}{\Delta t}\\ I^{\prime }=\underset{\Delta t\to 0}{\lim}\frac{I_{t+1}-I_{t}}{\Delta t}\\ R^{\prime }=\underset{\Delta t\to 0}{\lim}\frac{R_{t+1}-R_{t}}{\Delta t} \end{array}$$

Further,

(13)$$\{\begin{array}{c} S_{t+1}=S_{t}+\Delta tS^{\prime }\\ I_{t+1}=I_{t}+\Delta tI^{\prime }\\ R_{t+1}=R_{t}+\Delta tR^{\prime } \end{array}$$

According to Figure 5, the differential equations can be obtained, as follows:(14)$$\{\begin{array}{l} \frac{dS}{dt}=(1-p)\Lambda -(\mu +\eta +f(I))S+\gamma_{1}I+\delta R\\ \frac{dI}{dt}=f(I)S-(\mu +\eta +\varepsilon +\gamma_{1}+\gamma_{2})I\\ \frac{dR}{dt}=p\Lambda +\gamma_{2}I-(\mu +\eta +\delta )R \end{array}$$

Combining Equation (13) and Equation (14), the discrete-time SIRS model is as follows:(15)$$\{\begin{array}{l} S_{t+1}=S_{t}+\Delta t[\Lambda -(\mu +\eta +f(I))S+\gamma_{1}I+\delta R]\\ I_{t+1}=I_{t}+\Delta t[f(I)S-(\mu +\eta +\varepsilon +\gamma_{1}+\gamma_{2})I]\\ R_{t+1}=R_{t}+\Delta t[\gamma_{2}I-(\mu +\eta +\delta )R] \end{array}$$

Where f(I) is a real local Lipschitz function on set $R^{+}=\left[0,+\infty \right)$. (i) f(0)=0, and f(I)>0 when I>0; (ii) $\frac{f(I)}{I}$ is continuous and monotonously non-increasing when I>0, and $\underset{I\to 0^{+}}{\lim}\frac{f(I)}{I}=\beta$.

### 3.2. The Calculation of Information Entropy

The founder of information theory Shannon first put up with the measurement of information in 1948. He combined probability and statistics and took entropy as a measure of uncertainty or information regarding a stochastic event.

The entropy of a random variable X is defined as:(16)$$H(X)=H(p_{1},p_{2},\ldots ,p_{n})=-k\sum_{i=1}^{n}p_{i}\log_{2}p_{i}$$

In Equation (16), k is constant and k≥0. $p_{i}$ represents the probability that the system is in the $i^{th}$ microstate. H(X) will reach its maximum value when all states are equiprobable, that is, $p_{1}=p_{2}=\ldots =p_{n}=\frac{1}{n}$. The entropy will equal to zero if entire information is available. Otherwise, the entropy is greater than zero. For example, if each $p_{i}=1$ in a randomized trial X, then H=0.

In the whole assembly process, the entropy of the critical point is

(17)$$H_{c}=-\frac{O_{output}}{I_{input}}\log_{2}\frac{O_{output}}{I_{input}}=-\frac{S_{convergence}+R_{convergence}}{S_{initial}+I_{initial}+R_{initial}}\log_{2}\frac{S_{convergence}+R_{convergence}}{S_{initial}+I_{initial}+R_{initial}}$$

At the initial time, the entropy of whole process is

(18)$$H_{I}=0$$

Therefore, the entropy increment in assembly process is 

(19)$$\Delta H_{base}=H_{c}-H_{I}$$

The entropy increment of optimal assembly approach is

(20)$$\Delta H=-\sum_{i=1}^{m}\frac{O_{outputi}}{I_{inputi}}\log_{2}\frac{O_{outputi}}{I_{inputi}}$$

The entropy increment and the benchmark entropy are available through the above analysis and calculation. If the real-time entropy increment ΔH is greater than $\Delta H_{base}$, the defects emergence will happen during the assembly process. If the $\Delta H<\Delta H_{base}$, we need to calculate the probability of defects emergence. Additionally, the equation is shown, as follows:(21)$$P=\frac{\Delta H}{\Delta H_{base}}\times 100\%$$

## 4. Case Study

### 4.1. Case Introduction

In this section, a practical example is illustrated for the method proposed. The oil pump is a device that transfers liquid from one chamber to another chamber isolated from it through some form of mechanism movement to realize the change of volume. When the discharged liquid encounters resistance, a certain liquid pressure is established between its outlet and liquid resistance. It is an energy conversion device that converts mechanical energy into hydraulic energy. It forms the power source of servo system together with the prime mover and it plays an important role in the servo system. Its structure is complex and requires high machining accuracy. 

According to production schedule, we need to assemble and adjust a certain type of oil pump. The explosive view of this type oil pump is shown in Figure 6. While considering the existing assembly conditions, we decided to adopt two identical production lines for assembly and adjustment after thinking twice. It is assumed that the reliability of each assembly process of this production line is 0.95. According to the production plan, 360 pieces of oil pump shall be assembled, and every 30 pieces of pump shall be packaged for transportation. There are two positions for the process inspection station while considering the actual working condition and technical route. There are three rework paths corresponding to position 1 and two rework paths corresponding to position 2 just as Figure 7 and Figure 8 shows, which means that there are five ways to assemble the product. Therefore, it is necessary to optimize the location of inspection station and rework path to ensure the maximum reliability and lowest cost of the assembly system.

### 4.2. Optimal Assembly Approach Selection

Table 2 shows the process capacity of assembly. Through calculation, the maximum output is $O_{1\max}=O_{2\max}=220.528$. There are three combinations of production, and they are $D_{1}=(210,150)$, $D_{2}=(180,180)$, and $D_{3}=(150,210)$, respectively.

For rework path 1 corresponding to position 1, the sequence of assembly process is shown as in Table 3.

(1) ASSEMBLING ACCORDING TO D1

The input loading for line 1 is $I_{1}=302.874$ and the calculation result is shown in Table 4

The input loading for line 2 is $I_{2}=216.339$ and the calculation result is shown in Table 5.

According to Table 2, the minimum capacity vector is

$C_{1}=[350,300,300,280,260,260,250,250,250,250,250,200,200,200,170,200]$.

(2) ASSEMBLING ACCORDING TO D2

The input loading for line 1 is $I_{1}=259.606$ and the calculation result is shown in Table 6.

The input loading for line 2 is $I_{2}=259.606$ and the calculation result is shown in Table 7.

According to Table 2, the minimum capacity vector is

$C_{2}=[300,250,250,250,240,220,210,200,300,250,250,250,240,220,210,200]$.

(3) ASSEMBLING ACCORDING TO D3

The input loading for line 1 is $I_{1}=216.339$ and the calculation result is shown in Table 8.

The input loading for line 2 is $I_{2}=302.874$ and the calculation result is shown in Table 9.

According to Table 2, the minimum capacity vector is

$C_{3}=[250,250,250,200,200,200,170,200,350,300,300,280,260,260,250,250]$.

According to Equation (11), the reliability of assembly system is $R_{11}=0.6876$.

Similarly, input loading, minimum capacity vector, and reliability corresponding to other positions and other paths can be calculated. Table 10 shows the calculation results.

The reliability of rework 2 and rework 3 in position 1 are all the same as Figure 9 shows. In this case, the amount of raw materials consumed in the same product is carried as the criterion for evaluating assembly way, and the result is shown in Figure 10. It shows that the raw materials consumed in rework 3 are less than that in rework 2. Therefore, rework 2 is better than rework 3 under the same reliability.

Similarly, we can get the result that rework 1 in position 1 is better than rework 1 in position 2.

In conclusion, the rank of different assembly paths is shown Table 11. Therefore, when the inspection station is placed after process 6th and rework path 2 is chosen, the reliability of whole assemble system will be the highest.

Further, the repair rate is 

(22)$$\gamma =\frac{1}{m(n-k+1)}\sum_{i=k}^{n}\sum_{j=1}^{m}\frac{Q_{ji}(R)}{l_{ji}}=0.045$$

### 4.3. Risk Evaluation based on SIRS Model and Entropy

The discrete-time SIRS epidemic model is applied to research the variation of different product states. From the calculating results shown in Figure 11, we can see that the interesting bifurcation phenomenon breaks out with time passing by, which can explain why some defects will happen during the assembly process.

Figure 12 shows the information entropy from the initial time to critical time point. It shows that the entropy will converge to a constant value $H_{c}=0.3007$. The entropy increment is

(23)$$\Delta H_{base}=H_{c}-H_{I}=0.3007-0=0.3007$$

To the optimal assembly scheme, the entropy increment in the whole assemble process is shown in Equation (24). Therefore, the defect emergence will happen in the assemble process, which means the assemble risk is existing.

(24)$$\begin{array}{ll} \Delta H & =-\sum_{i=1}^{3}\frac{O_{outputi}}{I_{inputi}}\log_{2}\frac{O_{outputi}}{I_{inputi}}=-(\frac{360}{302.187+215.848}\log_{2}\frac{360}{302.187+215.848}\\ & +\frac{360}{259.018+259.018}\log_{2}\frac{360}{259.018+259.018}+\frac{360}{302.187+215.848}\log_{2}\frac{360}{302.187+215.848})\\ & =1.095\\ & >\Delta H_{base} \end{array}$$

## 5. Conclusions and Discussions

In this paper, a novel quantitative risk evaluation method for the emergence of defects in the assembly process was put forward. We built an assembly model considering multiple production lines to evaluate the performance of assembly system. The assembly system was modeled as an assembly network through the process paths and decomposition. Subsequently, we calculated the capacities that assembly system can meet the order’s requirement. The issue on decision-making by reliability and cost was also emphasized. Therefore, it is beneficial for decision maker to choose the optimal assembly way caused by the location of inspection station. The emergence phenomenon of defects during the assembly process can be explained by applying the SIRS epidemic model into the assembly process. The critical time when the defect happens can be described by the bifurcation of differential equations. Therefore, we can obtain the simulative entropy increment ($\Delta H_{base}$) according to the SIRS model. To the optimal assembly way, the actual entropy increment (ΔH) during the assembly process can also be calculated. The assembly risk that can be quantified can be assessed based on the degree of their proximity.

This model can assess the risk of defects emergence during the assembly process. We calculated the actual entropy increment ΔH in the rest assembly approaches, and the results are shown in Table 12. The criterion that we chose the assembly approach was reliability and cost in our research. However, the entropy increment of the optimal assembly approach is not the minimal, and the entropy increment in position 1 rework 3 is even smaller (ΔH=1.090). This is an interesting phenomenon. It would be one research direction for us to rethink the two ways and determine which assembly way is better on earth, and the evaluation method might be improved in the future.

The static structure complexity (optimal assembly approach selection) and dynamic process complexity (the variation based on SIRS model) are involved in this study. It is essential to ensure the validity of the control strategies at the same time. This part is not concerned and it will also be one of our research interests in the future.

## Figures and Tables

**Figure 1 entropy-21-01029-f001:**
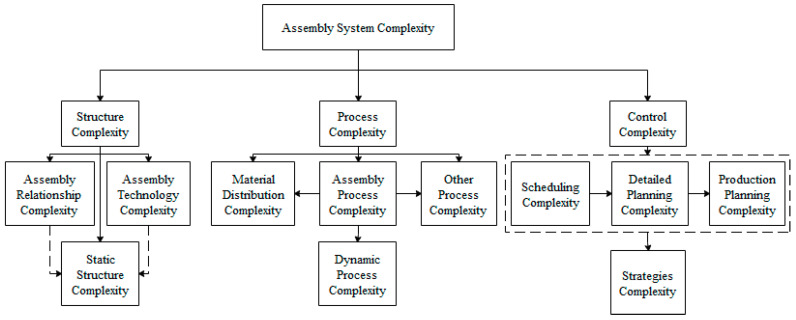
Classification of assembly system complexity.

**Figure 2 entropy-21-01029-f002:**
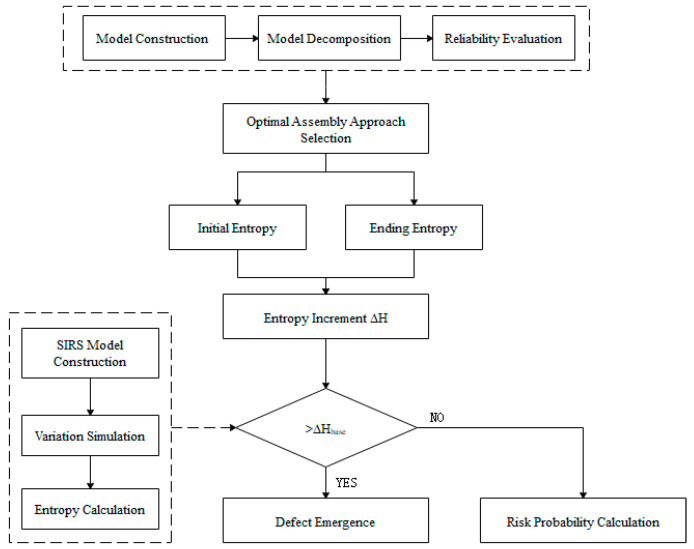
Framework for risk evaluation in assembly process.

**Figure 3 entropy-21-01029-f003:**
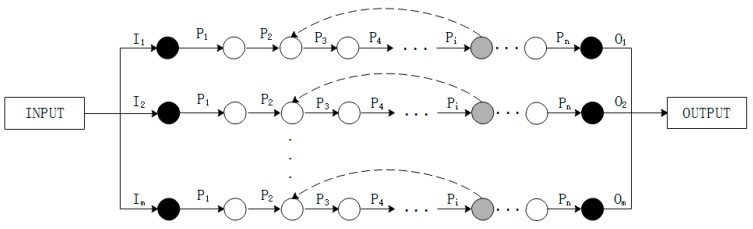
The model of assembly process.

**Figure 4 entropy-21-01029-f004:**
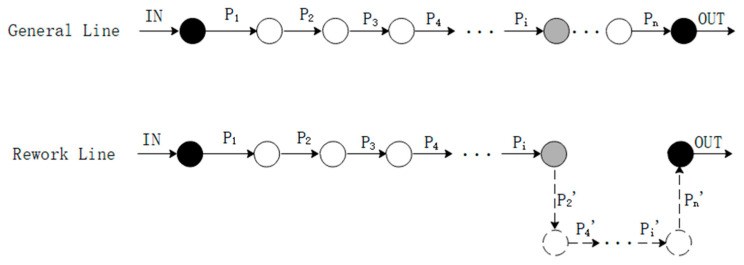
The decomposition model.

**Figure 5 entropy-21-01029-f005:**
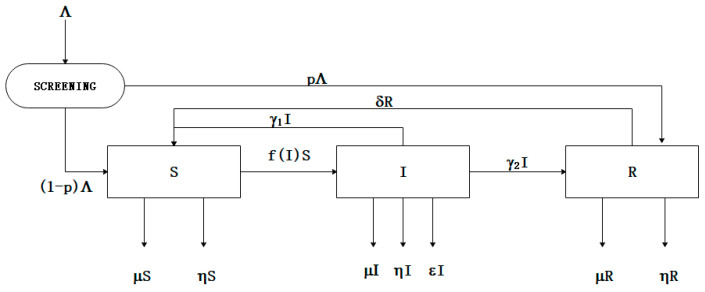
The SIRS model in assembly process considering screening [51].

**Figure 6 entropy-21-01029-f006:**
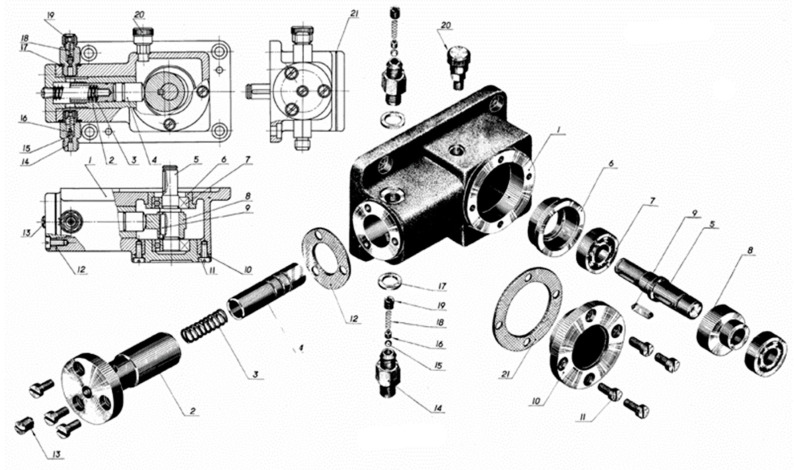
The explosive view of oil pump.

**Figure 7 entropy-21-01029-f007:**
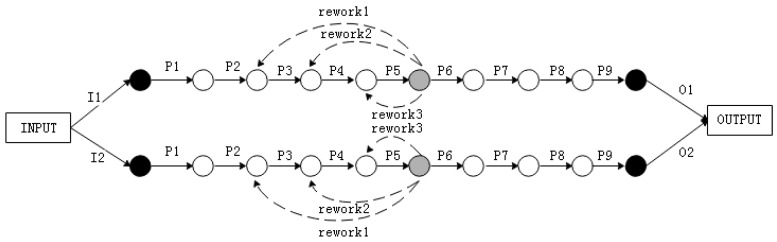
The assembly model based on position 1.

**Figure 8 entropy-21-01029-f008:**
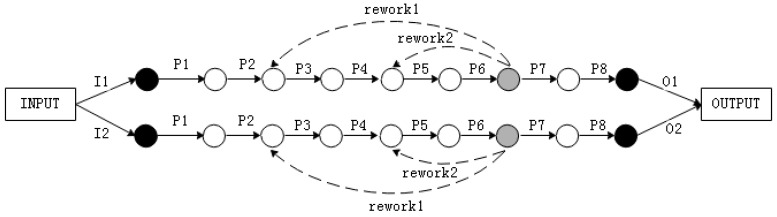
The assembly model based on position 2.

**Figure 9 entropy-21-01029-f009:**
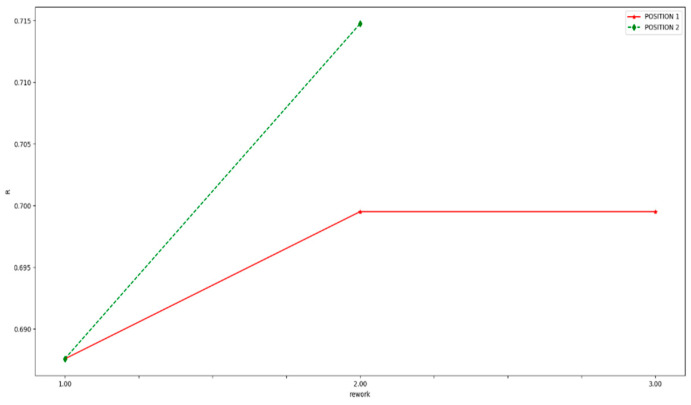
The reliability comparison of different assembly paths.

**Figure 10 entropy-21-01029-f010:**
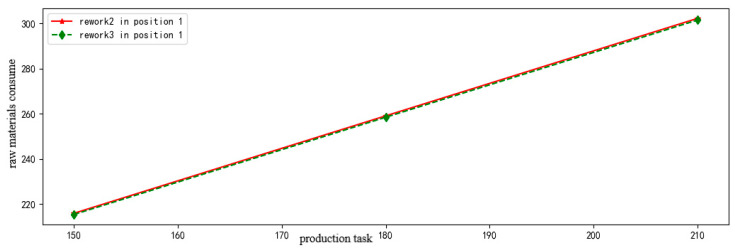
The raw materials consume comparison of different assembly paths.

**Figure 11 entropy-21-01029-f011:**
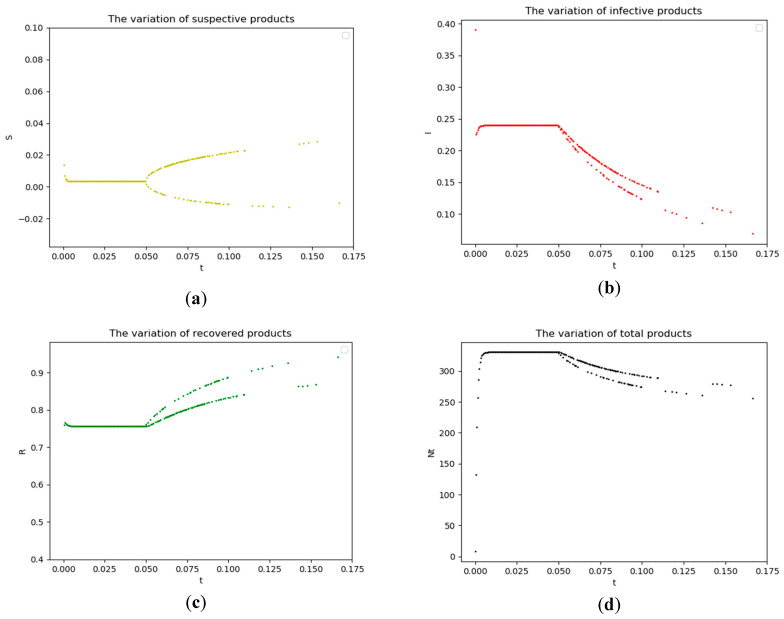
The variation of different states during assembly process. (**a**) The variation of susceptive products; (**b**) The variation of infective products; (**c**) The variation of recovered products; and, (**d**) The variation of total products.

**Figure 12 entropy-21-01029-f012:**
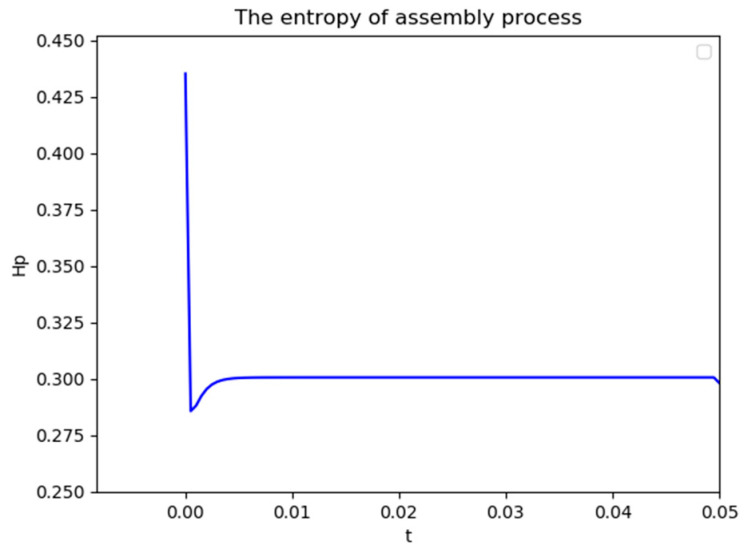
The entropy of assembly process.

**Table 1 entropy-21-01029-t001:** The comparison between defect emergence and disease propagation [52].

Definition	Infectious Source	Infectious Path	Susceptible Individuals
Disease propagation	Individuals with pathogens	The process that pathogens arrive and invade new susceptible individuals	Individuals susceptible to an infectious disease that lacks immunity or adaptive immunity
Defect emergence	Process that has a positive or negative effect on product defects under process stress	A time series process in which product defects are corrected or excited under stress	Hidden risk of infection or Lack of resilience to risks

**Table 2 entropy-21-01029-t002:** Process capacity in assembly.

Process	Capacity	Probability	Process	Capacity	Probability
P_1_	0	0.010	P_5_	0	0.001
	100	0.010		60	0.002
	150	0.010		120	0.002
	200	0.010		180	0.005
	250	0.010		200	0.010
	300	0.020		240	0.005
	350	0.020		260	0.005
	400	0.910		280	0.970
P_2_	0	0.005	P_6_	0	0.010
	150	0.010		50	0.010
	200	0.010		100	0.010
	250	0.015		150	0.020
	300	0.010		200	0.020
	350	0.010		220	0.005
	450	0.020		240	0.003
	600	0.920		260	0.001
P_3_	0	0.002	P_7_	280	0.001
	50	0.003		300	0.920
	100	0.005		0	0.005
	150	0.010		150	0.005
	200	0.010		170	0.015
	250	0.015		190	0.015
	300	0.955		210	0.015
P_4_	0	0.001	P_8_	230	0.025
	50	0.001		250	0.010
	100	0.001		270	0.910
	150	0.002		0	0.001
	200	0.002		50	0.002
	250	0.003		100	0.002
	280	0.010		150	0.005
	300	0.010		200	0.010
	320	0.010		250	0.015
	350	0.960		300	0.965

**Table 3 entropy-21-01029-t003:** Process sequence for assembly.

Process	$S_{ij}$	Following Processes	$t_{ij}$	$\alpha_{ij}$	$\beta_{ij}$
P_1_	1	P_2_, P_3_, P_4_, P_5_, P_6_, P_7_, P_8_	7	1	0
P_2_	2	P_3_, P_4_, P_5_, P_6_, P_7_, P_8_	6	1	0
P_3_	3	P_4_, P_5_, P_6_, P_7_, P_8_	5	0	1
P_4_	4	P_5_, P_6_, P_7_, P_8_	4	0	1
P_5_	5	P_6_, P_7_, P_8_	3	0	1
P_6_	6	P_7_, P_8_	2	0	1
P_7_	7	P_8_	1	0	1
P_8_	8	-	0	0	1

**Table 4 entropy-21-01029-t004:** Calculations for line 1 in D1.

	1	2	3	4	5	6	7	8
$Q_{i1}\left(G\right)$	302.874	287.73	273.344	259.677	246.693	234.358	222.64	211.508
$Q_{i1}\left(R\right)$	0	0	12.9838	12.3346	11.7179	11.132	10.5754	10.0466
$l_{i1}$	302.874	287.73	286.328	272.011	258.411	245.49	233.216	221.555

**Table 5 entropy-21-01029-t005:** Calculations for line 2 in D1.

	1	2	3	4	5	6	7	8
$Q_{i2}\left(G\right)$	216.339	205.522	195.246	185.483	176.209	167.399	159.029	151.077
$Q_{i2}\left(R\right)$	0	0	9.2747	8.81046	8.36994	7.95144	7.55387	7.17618
$l_{i2}$	216.339	205.522	204.52	194.294	184.579	175.35	166.583	158.254

**Table 6 entropy-21-01029-t006:** Calculations for line 1 in D2.

	1	2	3	4	5	6	7	8
$Q_{i1}\left(G\right)$	259.606	246.626	234.295	222.58	211.451	200.878	190.835	181.293
$Q_{i1}\left(R\right)$	0	0	11.129	10.5726	10.0439	9.54173	9.06464	8.61141
$l_{i1}$	259.606	246.626	245.424	233.153	221.495	210.42	199.899	189.904

**Table 7 entropy-21-01029-t007:** Calculations for line 2 in D2.

	1	2	3	4	5	6	7	8
$Q_{i2}\left(G\right)$	259.606	246.626	234.295	222.58	211.451	200.878	190.835	181.293
$Q_{i2}\left(R\right)$	0	0	11.129	10.5726	10.0439	9.54173	9.06464	8.61141
$l_{i2}$	259.606	246.626	245.424	233.153	221.495	210.42	199.899	189.904

**Table 8 entropy-21-01029-t008:** Calculations for line 1 in D3.

	1	2	3	4	5	6	7	8
$Q_{i1}\left(G\right)$	216.339	205.522	195.246	185.483	176.209	167.399	159.029	151.077
$Q_{i1}\left(R\right)$	0	0	9.2747	8.81046	8.36994	7.95144	7.55387	7.17618
$l_{i1}$	216.339	205.522	204.52	194.294	184.579	175.35	166.583	158.254

**Table 9 entropy-21-01029-t009:** Calculations for line 2 in D3.

	1	2	3	4	5	6	7	8
$Q_{i2}\left(G\right)$	302.874	287.73	273.344	259.677	246.693	234.358	222.64	211.508
$Q_{i2}\left(R\right)$	0	0	12.9838	12.3346	11.7179	11.132	10..5754	10.0466
$l_{i2}$	302.874	287.73	286.328	272.011	258.411	245.49	233.216	221.555

**Table 10 entropy-21-01029-t010:** Calculation results.

Position	Path	D	Input loading	Minimum capacity vector	Reliability
Position1	Rework 1	(210,150)	302.874	[350,300,300,280,260,260,250,250, 250,250,250,200,200,200,170,200]	0.68757846320055
			216.339		
		(180,180)	259.606	[300,250,250,250,240,220,210,200, 300,250,250,250,240,220,210,200]	
			259.606		
		(150,210)	216.339	[250,250,250,200,200,200,170,200, 350,300,300,280,260,260,250,250]	
			302.874		
	Rework 2	(210,150)	302.187	[350,300,300,280,260,260,250,250, 250,250,200,200,200,200,170,200]	0.69951463495432
			215.848		
		(180,180)	259.018	[300,250,250,250,240,220,210,200, 300,250,250,250,240,220,210,200]	
			259.018		
		(150,210)	215.848	[250,250,200,200,200,200,170,200, 350,300,300,280,260,260,250,250]	
			302.187		
	Rework 3	(210,150)	301.468	[350,300,300,280,260,260,250,250, 250,250,200,200,200,200,170,200]	0.69951463495432
			215.334		
		(180,180)	258.401	[300,250,250,250,240,220,210,200, 300,250,250,250,240,220,210,200]	
			258.401		
		(150,210)	215.334	[250,250,200,200,200,200,170,200, 350,300,300,280,260,260,250,250]	
			301.468		
Position2	Rework 1	(210,150)	303.529	[350,300,300,280,260,260,250,250, 250,250,250,200,200,200,170,200]	0.68757846320055
			216.807		
		(180,180)	260.168	[300,250,250,250,240,220,210,200, 300,250,250,250,240,220,210,200]	
			260.168		
		(150,210)	216.807	[250,250,250,200,200,200,170,200, 350,300,300,280,260,260,250,250]	
			303.529		
	Rework 2	(210,150)	302.187	[350,300,200,280,260,260,250,250, 250,250,200,200,200,200,170,200]	0.71474526994618
			215.848		
		(180,180)	259.018	[300,250,250,250,240,220,210,200, 300,250,250,250,240,220,210,200]	
			259.018		
		(150,210)	215.848	[250,250,200,200,200,200,170,200, 350,300,200,280,260,260,250,250]	
			302.187		

**Table 11 entropy-21-01029-t011:** Reliability ranking.

	Path	Reliability	Rank
Position 1	Rework 1	0.687578	4
	Rework 2	0.699514	3
	Rework 3	0.699514	2
Position 2	Rework 1	0.687578	5
	Rework 2	0.714745	1

**Table 12 entropy-21-01029-t012:** The actual entropy increment of assembly approaches.

Position	Rework path	ΔH
1	1	1.099
	2	1.095
	3	1.090
2	1	1.103
	2	1.095

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
