# Peer review of "The Method for Risk Evaluation in Assembly Process based on the Discrete-Time SIRS Epidemic Model and Information Entropy"

_entropy, 2019, doi:10.3390/e21111029_

Round 1
Reviewer 1 Report
It is an interesting research topic by employing epidemic model in risk assessment of assembly processes. It is a novel idea and I think it will be promising.
The authors mentioned that ‘We have had a comparison from the aspects of infectious source, infectious path and susceptible individuals. We apply the SIRS epidemic model to simulate the critical entropy during the assembly process when defects emerge’. I guess it is in their previous research ref [40]. It will be good to see the comparison here in this paper as it is the foundation of the research. The ref is a conference paper, which is not easy to access.
The authors proposed the method of ‘Optimal assembly path selection based on reliability and cost’, which is another perspective of the paper. There is little literature on this topic and it was not mentioned in the conclusions.
The conclusion is weak. It is not clear how effective the proposed risk evaluation method is.
Reviewer 2 Report
The presented paper deals with a method to assess the risk in assembly process based on discrete-time SIRS model. The method si well developed in details. However the structure of the paper could benefit from a revision. the section 2 describes the notations and assumptions which could be added in the beginning of the paper. some others sections are too short and should combined with the others (section 4.3 should be added in 3.3 for example). the case study is well detailed and the numerical results are described good.
